# Antibiotic Considerations in the Treatment of Maternal Sepsis

**DOI:** 10.3390/antibiotics14040387

**Published:** 2025-04-07

**Authors:** Sarah E. White, R. Phillips Heine, Talla M. Widelock

**Affiliations:** Department of Obstetrics Gynecology, Section of Maternal Fetal Medicine, Wake Forest University School of Medicine, Winston-Salem, NC 27157, USA; rheine@wakehealth.edu (R.P.H.); twideloc@wakehealth.edu (T.M.W.)

**Keywords:** maternal sepsis, sepsis, pregnancy physiology, antibiotics

## Abstract

Sepsis, a life-threatening organ dysfunction caused by a dysregulated host response to infection, remains the third leading cause of maternal mortality globally. Pregnancy-associated physiological adaptations predispose pregnant individuals to infection, impair maternal response to infection, affect antibiotic pharmacokinetics and metabolism, and complicate diagnosing infections and sepsis. Therefore, it is tantamount that clinicians readily recognize maternal sepsis and understand antibiotic regimens and treatment principles to avoid adverse maternal outcomes. In this article, we present an overview of the diagnosis and management of maternal sepsis and the physiological changes in pregnancy that alter antibiotic pharmacokinetics. Common microorganisms implicated in maternal sepsis are discussed with an emphasis on *E. coli* and Group A *Streptococcus* due to their prevalence and morbidity in the pregnant population. Lastly, we provide an overview of commonly used antibiotics and dosage recommendations in the treatment of maternal infection and sepsis.

## 1. Introduction

Sepsis is defined as life-threatening organ dysfunction caused by a dysregulated host response to infection. Maternal sepsis specifically is defined by the World Health Organization as a life-threatening condition with organ dysfunction resulting from infection during pregnancy, childbirth, postabortion, or up to 42 days postpartum [1]. Sepsis is the third leading cause of maternal mortality worldwide, resulting in 11% of maternal mortality globally [2]. Despite increasing awareness of and improvements in the care of maternal sepsis, the proportion of maternal deaths related to maternal sepsis has remained relatively unchanged since the 1980s [3]. More contemporary data from the US Nationwide Inpatient Sample demonstrated an annual 10% rise in maternal deaths between 1998 and 2008 [4]. This concerning trend has been suggested to be secondary to several factors, including increased antibiotic resistance, maternal age, co-morbidities, and microbiological factors such as an increased incidence of *Escherichia coli* and Group A streptococcal infections [5].

Pregnancy and the peripartum state are associated with various normal physiologic and immunologic adaptations (see Table 1). While these adaptations serve to protect the fetus, they also predispose peripartum patients to infection and severe infectious morbidity, and they obscure the clinical presentation, making the diagnosis of maternal sepsis challenging [6,7,8].

Pregnancy-specific risk factors for maternal sepsis include asymptomatic bacteriuria or acute cystitis, a history of/current carriage status of Group B streptococcus, operative interventions in the peripartum period, prolonged labor or rupture of membranes, and Group A streptococcus carriage or infection in close contacts. Additionally, there are various patient-related factors or co-morbidities which contribute to the risk of maternal infection and sepsis. These factors for maternal sepsis include primiparity, pre-existing medical conditions (e.g., obesity, renal disease, diabetes, heart disease, and autoimmune conditions such as SLE), ethnic minority status, health inequities, and socioeconomic deprivation [4,9,10]. As these medical factors become more prevalent in the obstetric population, it becomes increasingly salient to maintain clinical vigilance and expertise in the diagnosis and management of sepsis in pregnancy.

## 2. Sepsis and Septic Shock

In 2016, the Third International Consensus Definitions for Sepsis and Septic Shock Task Force recommended the adoption of the terms sepsis and septic shock along with the use of the Sequential Organ Failure Assessment (SOFA) score to identify organ failure [11]. Sepsis is denoted by an increase in the SOFA score of 2 or more and is associated with an in-hospital mortality rate of 10%. Septic shock is characterized by a vasopressor requirement to maintain a mean arterial pressure of 65 mm Hg or greater and serum lactate level >2 mmol/L in the absence of hypovolemia; septic shock is a subset of sepsis in which mortality is substantially increased [11]. To more rapidly identify patients with suspected infection at risk of sepsis outside of the intensive care setting, the quickSOFA (qSOFA) score has been recommended. The qSOFA score evaluates the presence of three clinical criteria: systolic blood pressure ≤ 100 mm Hg, respiratory rate ≥ 22 per minute, and altered mental status. The presence of two or more of these factors places a patient at increased risk for poor sepsis-related outcomes [11,12]. However, it should be noted that the qSOFA score is not used to define sepsis but rather to identify patients at high risk of developing adverse outcomes.

Diagnosis of maternal sepsis can prove challenging as signs and symptoms of sepsis often mimic normal physiologic changes in pregnancy; none of the existing definitions of sepsis or septic shock account for these changes in pregnancy, and therefore, overdiagnosis or underdiagnosis of sepsis may occur [12]. In particular, the SOFA score assigns a point when an individual’s creatinine reaches 1.2 mg/dL, but this value is well above the normal range in pregnancy. Likewise, the SOFA score considers a mean arterial pressure (MAP) > 70 abnormal, but this value may be normal in the second and third trimesters [12]. Similarly, a platelet count < 150 × 10^3^/µL confers points within the SOFA score, whereas this can be a normal finding consistent with gestational thrombocytopenia in pregnancy.

In order to account for these physiological changes and to help better identify pregnant individuals at risk of sepsis, several obstetric scoring systems have been proposed, though each has its limitations. Major early warning systems for maternal sepsis include the modified early obstetric warning system (MEOWS), the Maternal Early Warning Criteria (MEWC), and The Maternal Early Warning Trigger (MEWT) [13]. All three of these systems use a combination of vital sign parameters in conjunction with clinical status to identify patients at high risk of impending critical illness, and, similar to the qSOFA, none of these tools specifically define sepsis within a patient [13]. Despite the recognition of the need for early warning systems, the best tool for identifying sepsis or predicting mortality in pregnant individuals remains unclear, and most of these remain unvalidated for use in pregnancy. In a study comparing six modified obstetric early warning systems to a non-obstetric early warning system, the modified obstetric systems did not outperform the non-obstetric systems in identifying pregnant individuals at the highest risk for developing sepsis or subsequent adverse outcomes. Thus, a step-wise approach in which a simple bedside screening tool is followed by further evaluation for organ dysfunction is recommended for any patient at risk of sepsis, and it is imperative that clinicians have a low index of suspicion for sepsis [14].

## 3. Management of Sepsis

First published in 2004, The Surviving Sepsis Campaign has provided guidelines for the management of sepsis with updates published in 2008, 2012, 2017, and most recently, in 2021 [15,16]. Early identification and initiation of treatment is the crux of sepsis management. In the nonpregnant population, initiation of antimicrobial therapy is associated with a 79.9% survival to hospital discharge; conversely, each hour delay in initiation of antibiotics is associated with a 7.6% reduction in survival [17]. While initial studies on this golden hour excluded pregnant individuals, multiple studies have demonstrated similar outcomes in obstetric populations [18,19].

The initial assessment by a provider should include vitals, a focused history, current or prior infections, current or prior antimicrobial therapies, recent exposures or procedures, and a focused physical examination. If the fetus is viable, fetal monitoring may be considered if delivery or intervention on fetal behalf would be considered. Once the patient is stabilized, a more comprehensive history and physical may be performed, and rapid maternal delivery for fetal benefit can be considered [14,17].

Suppose the patient demonstrates signs or symptoms of hemodynamic instability (mean arterial pressure < 65 mm Hg, respiratory rate ≥ 25, shortness of breath, abnormal heart rate, chest pain, altered mental status, or cyanosis) or if the provider deems the patient unstable, rapid escalation of care should be initiated. At many institutions, this may be performed through a rapid response team, but if this is not available, escalation of care should not be delayed [14,17].

Following the initial assessment, those with concern for sepsis should have laboratory evaluation completed to include a complete blood count with differential, comprehensive metabolic panel, serum lactate level, and blood cultures; additional tests, such as urine or sputum cultures, viral testing, coagulation studies, and arterial or peripheral blood gases may be obtained as deemed clinically necessary. Empiric broad-spectrum antibiotics targeted at the suspected source of infection should also be initiated within one hour, with the plan to tailor the regimen once a definitive diagnosis is made. It is crucial not to delay antibiotics more than one hour if cultures cannot be obtained, and it is still recommended to obtain cultures even if antibiotics have been initiated [14,17].

While the Surviving Sepsis Campaign guidelines recommend giving at least 30 mL/kg crystalloid fluid within the first three hours after recognition of sepsis, a more restrictive approach is recommended in pregnant patients due to the risk of left ventricular diastolic dysfunction and resultant pulmonary edema and higher mortality rates. In pregnant individuals, early intravenous administration of 1–2 L crystalloid fluid is recommended with ongoing maintenance fluids or consideration of vasopressors if deemed clinically necessary to maintain a MAP of 65 mm Hg [14].

## 4. Etiology of Sepsis

Both obstetric and non-obstetric causes of maternal sepsis often relate to the timing of infection within the pregnancy or postpartum period. Genitourinary tract infections are the most common source of infection and maternal sepsis within the prenatal and postpartum periods, with pyelonephritis being the most common etiology of non-obstetric admissions in pregnancy [17]. Intra-amniotic infection is often implicated in intrapartum sepsis, whereas endomyometritis, mastitis, gastrointestinal, and soft tissue infections are common causes of infection and sepsis during the postpartum period. Common pathogens associated with maternal sepsis include *Escherichia coli*, group B streptococcus, *Staphylococcus aureus*, anaerobic bacteria, and *Listeria monocytogenes*. *E. coli* specifically is seen in up to one-half of positive blood cultures of pregnant individuals and is the primary bacteria seen in cases of urosepsis [17]. While less common, sepsis due to Group A streptococcus is associated with rapid clinical deterioration and has a mortality rate of 30–60% [20]. In the following section, we further discuss Group A Streptococcus and *Escherichia coli*, given their prevalence and association with increased maternal morbidity and mortality.

## 5. Septic Abortion and Postpartum Endomyometritis: Group A Streptococcus (GAS)

Septic abortion refers to an intra-uterine infection occurring at ≤20 weeks of gestational age. Following the Supreme Court decision in June 2022 of Dobb v. Jackson Women’s Health Organization [21], there has been concern for decreased access to safe abortions and increased abortion and pregnancy complications. Today’s clinicians should be knowledgeable regarding the recognition and treatment of septic abortion. In addition to the previously recommended goal-directed therapy, clinicians should assess for retained products of conception using ultrasonography. In patients suspected of having a more extensive infectious process, such as pelvic infection from uterine perforation, a computed tomography (CT) scan of the pelvis and abdomen should be considered. In septic patients with evidence of retained products of conception, early goal-directed therapy with intravenous fluids and antibiotics should be initiated. Uterine aspiration is recommended for source control, regardless of fetal cardiac activity.

Causative bacteria, in the case of septic abortion, generally arise from the vaginal flora; however, anaerobes should also be considered, especially in patients presenting with sepsis and shock. Although most septic abortion cases are believed to be polymicrobial, it is important to consider less common infectious etiologies in critically ill patients, especially those with septic abortion associated with postpartum endomyometritis. Timely diagnosis and targeted treatment are essential for improving patient outcomes. For example, endomyometritis may occasionally be caused by genital mycoplasmas or sexually transmitted organisms, such as *Chlamydia trachomatis*. In immunocompromised patients, such as those with HIV, other pathogens like herpes simplex virus and cytomegalovirus may be implicated. Additionally, *Group A streptococcus* (GAS) should be considered in critically ill postpartum or postabortion patients who present with septic shock, necrotizing fasciitis, or toxic shock syndrome.

Although relatively rare, the obstetric literature has reported a concerning rise in maternal mortality from sepsis in recent years, with an increase in Group A streptococcal genital tract infections contributing to this trend [3,19]. GAS surface proteins, including lipoteichoic acid and M proteins, enable the organism to invade intact epithelium and facilitate severe invasive diseases, such as necrotizing fasciitis, genital tract infections, or pneumonia [22]. Once GAS has invaded the epithelium, the hyaluronic acid capsule of the microbe promotes its multiplication, exacerbating host collapse [23]. Furthermore, the release of superantigens, such as streptococcal pyrogenic exotoxin, directly activates T cells, leading to massive cytokine production and, consequently, rapid host deterioration and multiple organ failure, as seen in streptococcal toxic shock syndrome [20]. In one report, 50% of pregnant and peripartum patients with proven Group A streptococcal infections deteriorated to septic shock more rapidly than those with other infectious organisms [9]. Due to the unique characteristics of GAS infections, mortality rates range from 30% to 60%. While broad-spectrum antibiotics can be used in the treatment of GAS infections, source control via hysterectomy may be needed for definitive treatment of patients who do not respond to antibiotics or deteriorate while receiving treatment. Awareness of this microbe and vigilance in maternal surveillance and treatment are crucial to reducing maternal morbidity and mortality [20,24].

## 6. Urinary Tract Infections: *Escherichia coli* (*E. coli*)

In the following section, we review *Eschereria coli* (*E. coli*) due to the frequency with which this microbe is implicated in maternal sepsis. *E. coli* is the most common organism identified on culture during the antenatal period and is the leading pathogen in all cases of maternal sepsis [25]. In one study, the most common cause of maternal sepsis in the third trimester of pregnancy was *E. coli* urinary tract infections, followed by genital tract infections [26]. Urinary tract infections (which include asymptomatic bacteriuria, acute cystitis, and pyelonephritis) have been estimated to affect approximately 2–15% of all pregnancies [27]. Overall, *E. coli* is the most common pathogen isolated from urine culture in pregnant patients, accounting for approximately 70–80% of positive cultures [28].

Historically, antibiotic therapy has been successful in the treatment of urinary tract infections; however, high recurrence rates and increasing antimicrobial resistance, including global emergence of multi-drug resistance among uropathogens (i.e., extended-spectrum B-lactamase/ESBL producing *E. coli*), may limit the efficacy of certain antibiotic regiments and limit antimicrobial options [29]. In these complex cases, it is prudent to remain vigilant for non-improvement or worsening maternal conditions and to consider ID consultation and changing antibiotic courses if necessary.

## 7. Pregnancy Physiology and Changes to Antibiotic Metabolism/Dosing

Several physiological changes that occur during pregnancy alter the pharmacokinetics of antibiotics, and these need to be accounted for when selecting and dosing antibiotics to treat pregnant individuals. These changes affect nearly all organ systems, begin early in gestation, and last up to 12 weeks postpartum; thus, it is imperative for providers to understand these changes to optimize the treatment of pregnant individuals and avoid morbidity [30].

Starting at 6–8 weeks gestation and peaking at 32 weeks, maternal plasma volume expands to volumes that are 40–50% higher than nonpregnant individuals. This is mirrored by increases in extracellular fluid space and total body water [31]. There is also a noted decrease in plasma albumin concentration, which, coupled with the increased plasma volume, leads to decreased colloidal pressure and hemodilution anemia. Due to the increased plasma volume and total body water, there is an increased volume of distribution for many hydrophilic drugs, and an increased initial and/or maintenance dose may be required to achieve therapeutic drug levels [30].

The gastrointestinal system also undergoes changes that alter the pharmacokinetics of medications in pregnancy. Driven by progesterone-mediated effects on smooth muscle, gastric emptying is delayed, and intestinal transit time is increased, which can affect the bioavailability of certain medications. Nausea and vomiting, which are commonly encountered in the first trimester, may also decrease drug absorption and plasma drug concentrations.

Decreases in serum albumin and alterations in maternal plasma pH decrease drug–protein binding and increase concentrations of the unbound drug. These changes in active drug or drug metabolite levels should be taken into consideration when dosing medications. Drug–drug interactions also should be considered as commonly taken medications in pregnancy, such as iron and antacids, may chelate other medications, further reducing plasma concentrations [30].

Hepatic drug-metabolizing enzyme activity is also altered in pregnancy due to the changes in estrogen and progesterone. Some enzymes, notably the CYP450 family, demonstrate increased activity in pregnancy and, therefore, increased the clearance of drugs, whereas other enzymes, such as xanthine oxidase and n-acetyltransferase, demonstrate decreased activity and decreased clearance of drugs [31].

In conclusion, physiological changes in pregnancy lead to an increase in glomerular filtration rate, an increase in total body volume, and enhanced cardiac output. Additionally, changes in the hepatic and gastrointestinal systems and metabolic pathways affect drug concentrations and clearance. These changes may lead to pharmacokinetic alterations in antibiotics that require dose adjustment or careful monitoring and assessment [32].

## 8. Antibiotic Regimens for Maternal Sepsis

Early and appropriate antibiotic treatment is crucial in managing maternal sepsis. Studies have shown that for every hour of delay in antibiotic administration after the onset of hypotension, survival decreases by 7.6% [33]. One of the leading and modifiable factors contributing to sepsis-related maternal deaths is the delay in providing appropriate treatment [5,32]. Therefore, prompt and accurate diagnosis, followed by the early initiation of antibiotics within one hour of hospital admission, is essential. This approach should focus on targeting the specific microorganisms most likely responsible for the infection to reduce the morbidity associated with maternal sepsis [34].

When initiating antimicrobial treatment for sepsis, the initial antibiotic choice is often empirical, based on clinical assessment, the suspected source of infection, local microbial resistance patterns, and hospital protocols. Thus, it is critical to be well-versed in the common infectious pathogens and their sources and etiologies, understand the characteristic clinical presentations and disease courses in pregnancy, and have a basic understanding of the treatment strategies for these infections in pregnancy (see Table 2). For peripartum sepsis specifically, empiric broad-spectrum antibiotic regimen options include the following: a broad-spectrum carbapenem (e.g., meropenem, imipenem/cilastatin, or doripenem); an extended-range penicillin/β-lactamase inhibitor combination (e.g., piperacillin/tazobactam or ticarcillin/clavulanate); or several third- or higher-generation cephalosporins, particularly as part of a multi-drug regimen (see Table 3) [16]. In following general sepsis guidelines, once a causative agent is identified, antibiotic regimens should be tailored to promote antibiotic stewardship.

The rationale for broad-spectrum antibiotic therapy is twofold: first, genitourinary tract infections, which are the most common source of maternal sepsis, are often polymicrobial and thus require broad-spectrum coverage; second, since *Group A Streptococcus* and *E. coli* are frequently associated with severe infections, initiating empirical antibiotics that cover gram-positive, gram-negative, and anaerobic pathogens is crucial. Additionally, if there is concern for *Group A Streptococcus*, antibiotic therapy should include clindamycin, as it inhibits exotoxin production by the pathogen and improves clinical outcomes when used alongside broad-spectrum antibiotics [35]. If clindamycin cannot be used, linezolid is an effective alternative for treating *Group A Streptococcus* infections [36].

There is relatively limited information regarding the microbial sources and etiologies of maternal sepsis. In the UK Obstetric Surveillance System, clinical laboratory testing identified the causative microorganism in only 64% of cases, the source was clinically identified in 74%, and neither the inciting organism nor the source was identified in 16% of maternal sepsis cases [9]. These statistics align with those observed in the general adult population; in one large study, blood cultures were negative in two-thirds of patients, and cultures from all sites were negative in one-third [37]. This, coupled with the fact that culture results typically take at least 24 h to become available, can make confirmation of a systemic microbial infection challenging in clinical practice [38]. New technologies, such as multiplex polymerase chain reaction (PCR) to identify multiple pathogen DNA sequences and their resistance genes, as well as the use of biomarkers like CRP, procalcitonin, and lactate to track disease progression, have been evaluated. There is cautious optimism that these advances may enable more timely and precise treatment, reduce ambiguity in antibiotic selection, and optimize therapy regimens [7,39,40].

When microbial pathogens are identified in maternal sepsis cases, the most common offenders include *Escherichia coli*, *Streptococcus* spp., *Staphylococcus* spp., other gram-negative bacteria, and mixed infections [41]. The issue of polymicrobial infection is significant, as the literature has shown that in 15% of maternal sepsis deaths where organisms were identified, the infection was polymicrobial [19]. As a result, initial antimicrobial therapy for maternal sepsis should generally be broad-spectrum, covering both anaerobic and aerobic gram-positive and gram-negative bacteria, with a focus on the most common pathogens: *E. coli*, *Staphylococcus* spp., *Streptococcus* spp., and other gram-negative bacteria. Viral and fungal coverage should be considered if suspected.

After initiating and selecting the appropriate antibiotic regimen for maternal sepsis, it is crucial to remain vigilant for signs of inadequate or ineffective treatment, which may present as rapid clinical deterioration or lack of improvement. In such cases, consulting with an infectious disease specialist should be considered, and alternative possible etiologies should be explored, along with optimization of source control. Generally, uncomplicated cases show clinical improvement within 24–48 h, and continued monitoring should be maintained to detect any signs of clinical worsening early. Although the optimal duration of antibiotic therapy remains a subject of debate and lacks high-quality evidence to guide practice, expert guidelines, including those from the Surviving Sepsis Campaign, suggest a 7–10-day course of antibiotics for treating sepsis. The duration can be adjusted based on the etiology and source of the infection, with consideration given to the potential need for suppression therapy or longer courses in certain cases [16].

As antibiotic resistance becomes an increasingly pressing issue, antibiotic stewardship is essential in addressing the growing morbidity and mortality associated with resistant pathogens. Effective antibiotic stewardship, particularly through the de-escalation of antibiotics and tailoring regimens to target specific microorganisms, is a cornerstone of sepsis management [16].

## 9. Specific Antimicrobials and Dosing Considerations in Pregnancy

The appropriate selection of antibiotics for the treatment of maternal sepsis is crucial as this has been shown to decrease mortality [17]. As previously discussed, once sepsis is suspected, broad-spectrum antibiotics should be promptly initiated, with the choice of antibiotic dependent on the suspected source of infection. However, in pregnant individuals, awareness of pregnancy’s effects on the antibiotic and how this alters recommended dosage recommendations is important to ensure adequate treatment. Table 2 provides a summary of information regarding conditions for which each antimicrobial regimen should be considered and common pathogens being targeted.

## 10. Beta-Lactams and Related Antibiotics

### 10.1. Penicillins

Penicillins and penicillin derivatives or combination drugs are the most widely prescribed antimicrobial class during pregnancy, with robust safety data [42,43]. During pregnancy, penicillins may have decreased serum concentrations by as much as 50% due to increased plasma volume and creatinine clearance. As such, potential dosage and frequency increases may be necessary [44]. Some penicillins, such as the anti-staphylococcal penicillins (except methicillin), demonstrate increased protein-binding, resulting in lower fetal tissue concentrations compared to low protein-binding penicillins (penicillin G or ampicillin) [44].

### 10.2. Cephalosporins

Cephalosporins have been used for a long time in pregnancy and remain a first-line option for many infections, particularly in cases in which the patient is allergic or intolerant to penicillin. First and second-generation cephalosporins have cross-reactivity rates of 1–8%, and third-generation cephalosporins have cross-reactivity rates of less than 1% [45]. During pregnancy, cephalosporins may have decreased plasma concentration due to increased renal elimination. Therefore, potential dosage and frequency increases may be required for some cephalosporins [46]. However, ceftriaxone specifically does not require dose adjustments in pregnancy, as studies have demonstrated that the serum concentration of standard doses of ceftriaxone remained greater than the minimum inhibitory concentration of ceftriaxone needed for causative organisms [47]. Studies have also shown that daily single-dose intravenous ceftriaxone is as effective as multiple-dose cefazolin in the treatment of patients with acute pyelonephritis in pregnancy [48].

## 11. Aminoglycosides

Gentamicin is the most commonly used aminoglycoside antibiotic in pregnancy. During pregnancy, aminoglycosides may have a lower serum peak concentration due to increased clearance and a larger volume of distribution [32,49]. Both traditional or extended interval dosing of aminoglycosides in pregnancy may be provided [50]. A single daily dose of 5 mg/kg actual body weight has been shown to be safe and clinically effective in maternal care and is recommended for patients with chorioamnionitis, postpartum endometritis, pyelonephritis, pelvic infection, and postoperative infections when an aminoglycoside is part of the recommended antibiotic regimen [50,51].

## 12. Carbapenems

Carbapenems such as ertapenem, meropenem, doripenem, and imipenem have been used safely and successfully for the treatment of infectious morbidities in pregnancy. While there is a relative paucity of data on carbapenem pharmacokinetics when utilized in pregnancy, there is theoretical potential that the physiologic changes of pregnancy may decrease concentrations; therefore, dosage and frequency increases may be required. Carbapenem therapy is typically reserved for pregnant patients with infections that are resistant to penicillin and cephalosporin therapy with limited alternatives [32,49,52].

## 13. Glycopeptides

Vancomycin is a glycopeptide antimicrobial drug used generally in the setting of serious gram-positive infections in pregnancy, such as methicillin-resistant *Staphylococcus aureus* (MRSA) infections [49]. Vancomycin crosses the placenta and is widely distributed in body tissues. While there are relative limitations to the quality of the safety data, the animal studies and case reports are reassuring and without reported associations with congenital malformations or fetal toxicity [49]. Vancomycin is mostly protein-bound and cleared by the renal system, both of which may be altered in pregnancy. However, population pharmacokinetic studies support the use of nonpregnant dosing in pregnant populations [53].

## 14. Macrolides

Macrolides, such as erythromycin, azithromycin, clarithromycin, or roxithromycin, historically are associated with disparate safety concerns in pregnancy despite having relatively low protein-binding, large volumes of distribution, and hepatic metabolism [46]. For example, erythromycin has been associated in some early reports with fetal cardiovascular malformations and pyloric stenosis, while later studies did not find an increased risk of congenital anomalies associated with macrolide use in pregnancy [54,55]. Conversely, azithromycin has generally been considered safe for use in pregnancy as animal and human studies have not demonstrated evidence of increased embryopathy, congenital anomalies, or fetal toxicity [49,56]. While not frequently used for the treatment of sepsis, macrolides may be used in the treatment of endomyometritis or severe pneumonia, and studies have demonstrated improved clinical outcomes and long-term mortality rates associated with macrolide use in severe pneumonia. For the treatment of sepsis, macrolides should be used in combination with other antibiotics rather than as a single agent, and standard dosing is recommended [57,58].

## 15. Linezolid

Linezolid is an oxazolidinone antimicrobial that is used for the treatment of gram-positive infections, including bacterial pneumonia, skin and soft tissue infections, and vancomycin-resistant enterococcal infections; it is most frequently used as an alternative to vancomycin in inpatient settings [59]. Linezolid use during pregnancy currently has a dearth of outcome data and pharmacokinetic data [60]. The available literature, though, is reassuring as linezolid appears to be well-tolerated by pregnant patients, and no serious adverse events have been reported [61]. While plasma concentrations of linezolid could be altered by pregnancy physiology, standard dosing is still recommended for pregnant individuals. However, for pregnant individuals with concomitant severe renal or hepatic impairment or obesity, blood level monitoring to maintain a serum trough concentration of 2–7 mg/L is recommended. Similarly, if a patient is not responding to treatment, monitoring blood levels and adjusting dosing is warranted to ensure an adequate dose is being provided [62].

## 16. Clindamycin

Clindamycin is a lincosamide antibiotic that is generally regarded as safe to use in pregnancy. It is frequently used to treat anaerobic infections, often in combination with gentamicin, but may also be used to treat methicillin-resistant *Staphylococcus aureus* infections and streptococcal infections [63]. Clindamycin crosses the placenta, is widely distributed into most body tissues, is highly plasma protein bound, and is cleared primarily by the renal system. Clindamycin pharmacokinetics do not appear to be affected by the physiological alterations of pregnancy. Therefore, dose adjustments are not required [64,65].

## 17. Metronidazole

Metronidazole is utilized as an anti-anaerobic antibiotic in cases of polymicrobial infection. It is most commonly used in cases of sepsis in which an intra-abdominal source is suspected, such as from the gastrointestinal tract or due to abdominal surgery; it is also used for severe anaerobic sepsis [66,67]. In many cases, metronidazole is included as part of a multi-drug regimen. However, the use of metronidazole for double anaerobic coverage is not recommended unless used for the treatment of *Clostridium difficile*, as data demonstrate similar efficacy as monotherapy but increased risk of microbial resistance and drug toxicity [68]. Metronidazole crosses the placenta; however, most studies have not shown an increased risk of congenital anomalies or other adverse fetal effects [69]. Metronidazole pharmacokinetics are not altered by the physiological changes of pregnancy; therefore, dose adjustments are not required [70].

## 18. Conclusions

The use of early, appropriate antibiotics is crucial in the management of maternal sepsis. The selection of an antibiotic regimen in pregnancy requires careful assessment of the clinical circumstances and consideration of the therapy risks and benefits, pharmacokinetics of drugs in the peripartum state, and principles of antibiotic stewardship in the face of emerging and growing challenges related to antibiotic-resistant pathogens. In general, we recommend choosing the appropriate antibiotic for the clinical situation regardless of pregnancy, with consideration given only to dose adjustment based on the antibiotic required.

## Figures and Tables

**Table 1 antibiotics-14-00387-t001:** Physiological Changes in Pregnancy and Sepsis.

	Pregnancy	Sepsis	Cumulative Effects
Cardiovascular	Decreased systemic vascular resistance Increased cardiac output Increased blood volume Increased heart rate	Decreased systemic vascular resistance Decreased blood pressure Increased heart rate Vasodilation	Increased risk of hemodynamic collapse
Respiratory	Increased minute ventilation Decreased functional residual capacity Increased tidal volume Baseline respiratory alkalosis	Increased pulmonary vascular permeability Increased respiratory rate Acute lung injury	Increased risk of pulmonary edema Increased risk of acute respiratory distress syndrome
Renal	Increased glomerular filtration rate Increased renal blood flow	Renal vasoconstriction Renal ischemia Cytokine-mediated renal cell injury	Acute renal injury
Coagulation	Increased Factor I, II, VII, VIII, IX, XII, plasminogen activator inhibitors I and II Decreased Protein S Unchanged anti-thrombin and Protein C	Increased procoagulant effects Increased thrombin production Decreased protein C fibrinolysis	Increased microvascular thrombus formation Decreased tissue perfusion End-organ dysfunction

**Table 2 antibiotics-14-00387-t002:** Etiology and treatment of common obstetric and non-obstetric sources of maternal infection.

Source	Infection	Likely Causative Pathogen(s)	Broad Spectrum Antibiotics by Suspected Source	Dosing Regimens
*Obstetrical*	Intra-amniotic infection	Polymicrobial	Ampicillin plus gentamicin * With cesarean delivery, add clindamycin or metronidazole for anaerobic coverage.	Ampicillin 2 g IV every 6 h Gentamicin 5 mg/kg IV once daily Clindamycin 900 mg IV every 8 h Metronidazole 500 mg IV
	Endometritis/Septic abortion	Polymicrobial	Ampicillin, gentamicin, and clindamycin or metronidazole. Alternative: metronidazole plus ceftriaxone	Ampicillin 2 g IV every 6 h Gentamicin 5 mg/kg IV once daily Clindamycin 900 mg IV every 8 h Metronidazole 500 mg IV Ceftriaxone 1–2 g IV once daily
	Skin/soft tissue infection	Polymicrobial	Vancomycin plus piperacillin-tazobactam. Concern for group A *Streptococcus* or *Clostridium perfrigens*: penicillin G plus clindamycin or linezolid.	Vancomycin 15 – 20 mg/kg actual body weight IV every 8–12 h (normal kidney function; monitoring required) Piperacillin-tazobactam 4.5 g every 6 h Penicillin G 4 million units every 4 h Linezolid 600 mg IV every 12 h
	Mastitis	*S. aureus*, *S. epidermidis* Streptococci species	Vancomycin plus piperacillin-tazobactam.	Vancomycin 15–20 mg/kg actual body weight IV every 8–12 h (normal kidney function; monitoring required) Piperacillin-tazobactam 3.375–4.5 g every 6 h
*Non-obstetrical*	Pyelonephritis	*E. coli*, *Klebsiella*, *Proteus*, gram positive organisms	Ceftriaxone Can consider monotherapy with ertapenem or piperacillin-tazobactam.	Ceftriaxone 1-2 g IV once daily Ertapenem 1 g IV once daily Piperacillin-tazobactam 3.375–4.5 g every 6 h
	Community acquired pneumonia	*S. pneumonia*, *K. pneumonian*, *H. influenza*, consider viral (influenza, COVID-19)	Cefotaxime, ceftriaxone, ertapenem, or ampicillin plus azithromycin	Cefotaxime 1–2 g IV every 6–8 h Azithromycin 500 mg IV once daily
	Hospital acquired pneumonia	*Pseudomonas*, *S. aureus*, *S. pneumonia*, *K. pneumonia*, *H. influenza*	Low-risk: ceftriazone, ampicillin-sulbactam, ertapenem, meropenem, imipenem, or cefepime. High mortality risks: include *Pseudomonas* (beta lactam plus aminoglycoside or quinolone) and MRSA coverage (vancomycin or linezolid).	Ampicillin-sulbactam 3 g IV every 6 h Cefepime 2 g IV every 8 h
	Appendicitis/Intra-abdominal	Polymicrobial	Single agent: doripenem, imipenem/cilastatin, meropenem, piperacillin/tazobactam Multi-agent: cefepime, ceftazidime plus metronidazole *Antibiotic monotherapy without surgery not recommended.*	Doripenem 500 mg IV every 8 h Imipenem/cilastatin 500 mg IV every 6 h or 1 g every 8 h Meropenem 1 g IV every 8 h Piperacillin-tazobactam 3.375–4.5 g IV every 6 h Cefepime 1 – 2 g IV every 8 – 12 h Ceftazadime 2 g IV every 8 h Metronidazole 500 mg IV every 8 h

* Gentamycin dosed every 24 hours, i.e. 5 mg/kg/day is preferred.

**Table 3 antibiotics-14-00387-t003:** Proposed broad-spectrum empiric antibiotic regimens of peripartum sepsis.

**1.** **Gram-negative and anaerobic coverage** Piperacillin/tazobactam 3.375 g IV q8h (extended infusion) or 4.5 g IV q6h **or** Meropenem 1 g IV q8h (if recent hospitalization or concern for multi-drug resistant organisms) **or** Cefepime 1–2 g IV q8h plus metronidazole 500 mg IV q8h **or** Aztreonam 2 g IV q8h (for women with severe penicillin allergy) plus metronidazole 500 mg IV q8h **or** Aztreonam 2g IV q8h plus clindamycin 900 mg IV q8h **PLUS** **2.** **Gram-positive coverage** Vancomycin 15−20 mg/kg q8h−q12h (goal trough 15–20 mcg/mL) **or** Linezolid 600 mg IV/PO q12h (for women with severe vancomycin allergy)

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
