# Peer review of "Antibiotic Considerations in the Treatment of Maternal Sepsis"

_antibiotics, 2025, doi:10.3390/antibiotics14040387_

Round 1

Reviewer 1 Report

Comments and Suggestions for Authors

The manuscript provides a well-structured and comprehensive review of maternal sepsis, highlighting antibiotic selection, physiological changes during pregnancy, and antimicrobial resistance challenges. The authors effectively present current clinical guidelines and best practices. The cited literature is up-to-date and well-selected, supporting the manuscript’s conclusions.

I recommend publication after minor revisions:

Lines 8–17: The introduction section within the abstract is slightly lengthy, while the discussion and conclusions are underdeveloped. Reduce the background and expand on the antibiotics discussed in the review. Clarify the key antimicrobial classes covered and their relevance to maternal sepsis.

Lines 25–26: An unnecessary paragraph break

Table 1: The abbreviation “ARDS” should be expanded

Line 57: Missing space before “mm Hg”

Table 2: Bacterial names should be italicized

Line 199: Missing “spp.” for Staphylococcus and Streptococcus

Figure 1: Convert to color format to improve clarity and make distinctions between categories clearer

Line 250: The court decision referenced needs a proper citation format

Lines 293 & 296: E. coli should be italicized

Lines 379–380: Missing dash in “Gram-positive”

Line 396: “streptococcal” should be lowercase and not italicized

Lines 401–408: Metronidazole discussion should address the common clinical practice of Double Anaerobic Coverage, which is not recommended (https://www.unmc.edu/intmed/_documents/id/asp/other-doubleanaerobiccoverage.pdf)

Author Response

Comment 1: Lines 8–17: The introduction section within the abstract is slightly lengthy, while the discussion and conclusions are underdeveloped. Reduce the background and expand on the antibiotics discussed in the review. Clarify the key antimicrobial classes covered and their relevance to maternal sepsis. Reply: The abstract has been revised to condense the first paragraph slightly and expand the discussion of what is covered in the article. 

Comment 2:  Lines 25–26: An unnecessary paragraph break. Reply: We have attempted to address this, but this appears to be a formatting issue with the journal rather than our document; I am unable to fix this currently but will bring this up to the editor.

Comment 3: Table 1: The abbreviation “ARDS” should be expanded. Reply: This has been addressed.

Comment 4: Line 57: Missing space before “mm Hg” Reply: This has been addressed. 

Comment 5: Table 2: Bacterial names should be italicized. Reply: This has been addressed. 

Comment 6: Line 199: Missing “spp.” for Staphylococcus and Streptococcus. Reply: This has been addressed. 

Comment 7: Figure 1: Convert to color format to improve clarity and make distinctions between categories clearer. Reply: This figure has been revised in color format. 

Comment 8: Line 250: The court decision referenced needs a proper citation format. Reply: This has been cited and added to our references.

Comment 9: Lines 293 & 296: E. coli should be italicized. Reply: This has been addressed.

Comment 10: Lines 379–380: Missing dash in “Gram-positive” Reply: This has been addressed.

Comment 11: Line 396: “streptococcal” should be lowercase and not italicized. Reply: This has been addressed.

Comment 12: Lines 401–408: Metronidazole discussion should address the common clinical practice of Double Anaerobic Coverage, which is not recommended. Reply: This has been added to the manuscript. 

Reviewer 2 Report

Comments and Suggestions for Authors

The authors present an overview work discussing clinical information on sepsis and septic shock, their management, and how these considerations should be adapted in cases of pregnancy. The manuscript is within the aim of the special issue but major revisions are necessary before further consideration. A main point is what is the added value of the work considering similar approaches. Some comments follow:

1) Sections 2 and 3 should be merged into one, as the primary focus should be maternal sepsis.
2) Figure 1 is adapted from another work; therefore, the necessary license for its use should be provided.
3) Lines 246–247 and 308–312 are unnecessary and should be deleted.
4) In Section 8, the authors provide information on antibiotics, but it is not always specifically related to their use in maternal sepsis. Instead, much of the content is general and well-known. Please revise this section to align with the aim of the work. 
5) Section 8 fit better with Section 5 and should be merged
6) A discussion on future directions, adjunctive therapies, or supportive care would be useful. Alternatively, a better organization of the sections within the text should be considered.

Author Response

Comment 1: Sections 2 and 3 should be merged into one, as the primary focus should be maternal sepsis. Reply: We feel that these two sections should remain separate as the focus on each section differs; combining the two sections would be the single section a bit unwieldy and more challenging to read. 

Comment 2: Figure 1 is adapted from another work; therefore, the necessary license for its use should be provided. Reply: This figure, while using information from another manuscript, was an original figure designed by one of the authors. The citation stating it was adopted from another manuscript has been removed. 

Comment 3: Lines 246-247 have been removed. 

Comments 4-6: 4) In Section 8, the authors provide information on antibiotics, but it is not always specifically related to their use in maternal sepsis. Instead, much of the content is general and well-known. Please revise this section to align with the aim of the work. 
5) Section 8 fit better with Section 5 and should be merged
6) A discussion on future directions, adjunctive therapies, or supportive care would be useful. Alternatively, a better organization of the sections within the text should be considered.

Reply: We substantially reorganized the manuscript to address the above comments. 

Round 2

Reviewer 2 Report

Comments and Suggestions for Authors

The authors have submitted an updated version of their manuscript, addressing some of the comments provided. While the manuscript has improved, there are still minor issues that need to be resolved.

Figure 1: The figure remains uninformative as it consists only of pie charts illustrating the frequency of bacterial classifications and the distribution of Gram-positive and Gram-negative bacterial isolates (50-50?). However, it lacks numerical data or additional info. So, is it original created based on what data? Because it shows distribution data. If it is based on literature please provide the relevant references. This issue needs to be clarified, taking into account the initial comment regarding the source and originality of the figure. Also, the resolution of the figure is very poor. 

Table 3: The authors focus on empirical regimens, but do not discuss them under the prism of evidence-based prescription guidelines. Why is this the case? A discussion on evidence-based approaches would provide a more comprehensive analysis.Or explain if the empiric approach is developed considering evidene-based approaches.

Sections 9–17: These sections could be merged under Section 8 and structured as sub-sections. The initial suggestion for a clearer and more logical organization of the manuscript should be reconsidered.

Author Response

Comment 1: Figure 1: The figure remains uninformative as it consists only of pie charts illustrating the frequency of bacterial classifications and the distribution of Gram-positive and Gram-negative bacterial isolates (50-50?). However, it lacks numerical data or additional info. So, is it original created based on what data? Because it shows distribution data. If it is based on literature please provide the relevant references. This issue needs to be clarified, taking into account the initial comment regarding the source and originality of the figure. Also, the resolution of the figure is very poor. 

Reply: It was unfortunate that the figure resolution did not translate from what we provided for the manuscript. We did add additional information to make the figure more informative, but we continued to note issues with resolution when transferring the image to various test documents. Thus, we have elected to remove this figure from the manuscript altogether. 

Comment 2: Table 3: The authors focus on empirical regimens, but do not discuss them under the prism of evidence-based prescription guidelines. Why is this the case? A discussion on evidence-based approaches would provide a more comprehensive analysis. Or explain if the empiric approach is developed considering evidene-based approaches.

Reply: We have reformatted a portion of the paper that discusses initial antibiotic selection for sepsis and included a clarifying statement that Table 3 specifies regimens for peripartum sepsis that should then be tailored once a causative agent is identified in accordance with general sepsis management guidelines. 

Comment 3: Sections 9–17: These sections could be merged under Section 8 and structured as sub-sections. The initial suggestion for a clearer and more logical organization of the manuscript should be reconsidered.

Reply: While we appreciate this feedback and did reformat a portion of the article during this round of revisions, we do not feel that completely reformatting the manuscript is necessary at this time.